# An explorative study with convenience vegetables in urban Nigeria—The Veg-on-Wheels intervention

**Harriette M. Snoek**[1]*, **Ireen Raaijmakers**[1], **Oluranti M. Lawal**[2], **Machiel J. Reinders**[1]

**1** Wageningen Economic Research, Wageningen University & Research, Wageningen, The Netherlands, **2** Department of Food Science and Technology, Federal University of Technology Akure (FUTA), Akure, Nigeria

\* harriette.snoek@wur.nl

**Data Availability Statement:** The data has now been shared under creative commons attribution 3.0 Netherlands license (cc-by) at 10.5281/zenodo.7096954.

## Abstract

Nigerian consumers have been found to view vegetables as healthy and health is a principal motivation for consumption; however, consumers also experience barriers related to preparation time and availability of vegetables. We therefore conducted a Veg-on-Wheels intervention, in which ready-to-cook, washed and pre-cut green leafy vegetables (GLV) were kept cool and sold for five weeks at convenient locations near workplaces and on the open market in Akure, Nigeria. Surveys were conducted prior to the intervention with 680 consumers and during the final week of the intervention with 596 consumers near workplaces and 204 consumers at the open market. Both buyers and non-buyers of the intervention were included; 49% buyers in the workplace sample and 47% in the open market sample. The Veg-on-Wheels intervention was successful, with high awareness, positive attitudes and high customer satisfaction. GLV intake was higher for Veg-on-Wheels buyers compared with non-buyers after the intervention, i.e., 10.8 vs. 8.0 portions per week, respectively. Also the intake of other vegetables was higher in the intervention group. The motives and barriers for buyers and non-buyers differed across the selling locations: main barriers were trust in the vendor and GLV source. These trust issues and vendor preferences were viewed as more important to respondents at the market than those near workplaces. This study is the first intervention study on the selling of ready-to-cook convenience vegetables in urban Nigeria. It shows that a market exists for convenience vegetables and that they have the potential to increase vegetable intake. Insights on both the food environment and consumers' motives and behaviour was crucial for designing and evaluating the intervention.

## Introduction

### The food environment as an entry point for interventions

The food systems approach is viewed as a useful way to improve diet quality [1]; lead to better, more sustainable health outcomes; and reach other United Nations Sustainable Development Goals (SDGs) [2]. The food system has been defined as all the elements and activities that relate

**Funding:** This research was part of the CGIAR-A4NH Flagship "Food Systems for Healthier Diets" funded by The Netherlands Ministry of Foreign Affairs and The Netherlands Ministry of Agriculture, Nature and Food Quality; https://www.government.nl/; grant WUR-KB22-003-001 (HS, IR, OL, MR). The funders had no role in study design, data collection and analysis, decision to publish, or preparation of the manuscript.

**Competing interests:** The authors have declared that no competing interests exist.

to the production, processing, distribution, preparation and consumption of food, and the output of these activities, including socio-economic and environmental outcomes [3]. The food environment, a component within the food system, is gaining increasingly policy attention and prominence in research in low- and middle-income countries (LMIC) as it has an important role in shaping diets, nutrition and health outcomes [1,3,4]. Food environments have been described as the interface where people interact with the wider food system to acquire and consume foods. This interaction encompasses external (availability, prices, vendor and product properties and marketing and regulation) and personal dimensions (accessibility, affordability, convenience and desirability) [5]. The food environment is asserted to be a critical entry point in the food system to implement interventions to support sustainable and healthy diets, as it contains the total scope of options that consumers use in forming their dietary behaviour, including buying, preparing and consuming [6]. Enhancing the food environment to improve healthy food intake is used as a starting point by several interventions. A systematic review on the influence of the urban food environment on nutrition and health outcomes in LMIC revealed that school and neighbourhood settings are the most studied parts of the food environment, followed by work and at home setting. Most of the included studies focussed on food environment characteristics availability and accessibility [7]. For example, research in Brazil found that increasing fruit and vegetables' availability may be an effective way to establish new healthy eating habits [8,9].

Mobile produce markets are one way to improve availability of and access to foods as was supported in pilot studies in low income communities in the US [10,11] and a review of mobile produce markets in the United States [12]. Similarly, mobile produce markets were found to be a promising strategy to improve access to fruit and vegetables in another systematic review study–and might even support healthy food purchasing and consumption [13]. However, more rigorous experimental designs are needed and also, few studies have been conducted in LMIC. Additionally, in the United Kingdom, the workplace environment is viewed as an ideal way to promote healthy dietary behaviours while targeting a large percentage of the population [14]. Vegetable availability and accessibility at or near the workplace may be a solution to overcome availability and convenience barriers, particularly when there are few nearby open markets, street vendors, specialized shops or supermarkets. Several studies conducted in urban Kenya found that when those who cook work formally outside the home, it negatively influences the consumption of green leafy vegetables (GLV), as he or she does not have the time to cook [15]. However, no interventions have been conducted in LMIC to increase vegetable availability at workplaces.

## Consumer behaviour in interactions with the food environment

Consumers make their food choices based upon what is available and accessible in the food environment (opportunities and constraints), their available time and resources (economic and social), and their personal characteristics–such as preferences, attitudes towards foods, knowledge, and other psychological factors [16]. Research on food choice provides insights in consumers' decision-making processes that underly dietary intake. Insights in these processes are essential to better understand and better respond to the relationship between changing food environment and consumer food choice [17]. Potential constraints against healthy food choices, including vegetable consumption in urban areas of Nigeria, include: limited year-round availability; affordability; the need for convenience; food safety issues; and the attraction to modern or Western lifestyles [18]. In line with this, availability and convenience of buying and preparing vegetables were the main barriers to vegetable consumption for urban Nigerian consumers [19]. This is also due to the few storage options at home and inadequate

preservation and transportation, leading to vegetables being wasted. The main motive for vegetable consumption was health [19]. Vegetables are believed to be medicinal, particularly in rural areas [20], and viewed as promising in the management of diseases (e.g., diarrhoea, stomach ailments, coughing, malaria, etc.). These results suggest that interventions to increase vegetable intake should overcome the main barriers of availability and convenience in preparation, while meeting the need for healthy foods. In addition to convenience and health, other motivations–such as price, mood and sensory aspects–also were reported as important motives for food choices [19]. In 2016, an intervention was implemented in Nigeria that used consumer behaviour as an entry point in the development of their intervention [21]. In their successful intervention 'Follow in My Green Food Steps', participants were motivated to change their cooking habits and recipes by adding additional GLV and a Knorr-brand, iron-fortified bouillon cube to their stews to improve iron intake. While the 'Follow in My Green Food Steps' intervention was focussed mostly on food handling practices at home, the current study combines an intervention in the food environment with insights from determinants of consumer food choice behaviour. In addition to motives and barriers, determinants related to ability (self-efficacy and perceived product inconvenience) were also included since in order to achieve behaviour change motivation, ability, and opportunity are needed [22].

## Importance of green leafy vegetables in the diet of urban Nigerians

There is ample scientific knowledge that adequate consumption of vegetables prevents several micronutrient deficiencies, helps maintaining health and reduces the risk of developing chronic diseases such as heart disease, high blood pressure, diabetes and several cancers [23–26]. In Nigeria, like in other LMIC worldwide, representative, individual-level data on food intake are scarce, limited to specific groups, and most often not up to date [1,23,27]. A nationwide dietary survey conducted in 2001–2003 showed that vegetable intake in Nigeria is below recommended levels [28]. A more recent meta-analysis showed that vegetable intake increased in sub-Saharan Africa over the past three decades but is lower for poorer compared to richer countries and for urban compared to rural populations [29]. While recent and representative data on vegetable consumption in Nigeria is lacking, vegetable eating habits have been reported and show that urban Nigerians eat a limited variety of vegetables. Tomatoes, onions, hot peppers and GLV are among urban residents' most frequently consumed vegetables [19] and are traditionally cooked in soups and/or stews and usually are eaten with a starchy staple (e.g., fufu, i.e., pounded yams). These soups and stews contribute greatly to macro- and micronutrient intake, including dietary fibre, protein (also due to the animal proteins added), vitamins, minerals and antioxidants [30]. GLV have the potential to reduce micronutrient deficiencies, as they are the main vegetable ingredient in these soups, and African indigenous GLV contain significant levels of micronutrients (i.e., vitamin A, vitamin C, folate, riboflavin, iron, zinc, calcium and magnesium) and dietary fibre, which are essential to human health [31].

## A quasi-experiment Veg-on-Wheels

In this paper, we present results from a pilot intervention study in Nigeria with mobile markets selling vegetables near workplaces. Although both mobile markets and interventions at workplaces have been found promising in increasing vegetable access and availability, no such intervention has been conducted in urban areas in LMIC. In this intervention, named Veg-on-Wheels, GLV were washed and pre-cut, kept cool and sold at convenient places near workplaces and times tailored to working hours. This paper's objective is twofold: (i) to provide insights on determinants of GLV purchase, storage, and consumption behaviour, and (ii) to

investigate whether the Veg-on-Wheels intervention may be an effective way to facilitate increased vegetable consumption. In addition, general evaluation measures of the intervention such as intervention awareness and satisfaction with the dishes were measured. A quasi-experimental design with control and intervention periods was used to quantitatively measure vegetable intake, intervention awareness and satisfaction, and determinants of intake (motives, barriers, self-efficacy). Unlike 'real experiments', such as randomised controlled trials, in quasi-experiments, researchers cannot randomly assign participants to 'treatment' and 'control' groups, thereby precluding the pure testing of cause and effect. Quasi-experiments commonly are used when it is not practical or ethical to randomise study participants into control and treatment conditions. Nevertheless, quasi-experiments have high external validity because they commonly take place in natural environments and, therefore, may help identify effective interventions to stimulate public health and provide a useful tool for policy making [32]. By using a quasi-experimental design, we also concur with very recent studies on stimulating vegetable consumption that also used quasi-experiments as an instrument for testing [e.g. 33,34].

## Materials and methods

### Veg-on-Wheels intervention

In the Veg-on-Wheels intervention, students from the Federal University of Technology Akure (FUTA) sold ready-to-eat GLV using bicycles and pushcarts that included cool boxes. The intervention ran for five weeks at the end of the wet season (November-December 2018). The Veg-on-Wheels vegetables were sold at four different points across Akure (the capital city of Ondo State, southwest Nigeria). Three sites were located near workplaces: the FUTA campus (FUTA) and state and federal secretariats (Secretariats). The fourth selling point was on the open market, as this is a principal purchase area for vegetables (the Oja Oba and Aule market). Amaranth and fluted pumpkin leaves were cleaned, washed, cut, packaged and kept cool prior to sale and sold during office working hours (roughly from 9 to 5; including the end of the working day when there is less availability in open markets). In this way, we aimed to decrease the time consumers had to spend on meal preparation, as well as decrease travel time and costs associated with purchasing fresh vegetables, as consumers who used Veg-on-Wheels did not need to travel from their workplaces to open markets, which usually are not near workplaces. To inform consumers about the health benefits of GLV, a leaflet, 'Why you must eat green leafy vegetables every day', was distributed around the selling locations by project team members who were not selling the vegetables, and were attached to the bicycles and pushcarts.

### Study procedure

Data for the control and intervention periods were collected through a paper-based, self-administered questionnaire with a trained interpreter present to clarify questions. The questionnaires were deployed separately, but at some distance from the vendors, and interpreters were wearing T-shirts with the Veg-on-Wheels logo on them to enhance awareness of their affiliation. An informed consent statement was read to the respondents after the selection questions to explain that the data would be handled confidentially, that respondents had the right to end the interview at any point without providing a reason and that the researchers from FUTA and WUR would be processing their anonymously provided answers. The respondents then were asked whether they agreed with this statement, and their answers were recorded. All respondents agreed and provided verbal consent, and none of the respondents was paid for their participation. Additional information regarding the ethical, cultural, and scientific considerations specific to inclusivity in global research is included in the supporting information "inclusivity questionnaire Snoek.docx".

### Respondent selection

For both the control and intervention periods, respondents were recruited in the vicinity of Veg-on-Wheels selling sites near workplaces and the FUTA and Secretariat sites. Questionnaires were administered in people's office when possible. A between-subjects design was used, since the research settings did not enable the option to match respondents reliably before and after the intervention, which is needed for a within-subjects comparison. Considering that initially, a within-subjects analysis was planned, efforts were made to find and interview the same respondents at the FUTA and Secretariat sites, and most respondents in the intervention sample (the sample collected during the intervention period) were the same as in the control sample (the sample collected during the control period). Additionally to the recruitment near FUTA and Secretariats sites, during the intervention period, an additional sample of respondents was recruited from the open market near the Veg-on-Wheels vendors.

Respondents were included only if they were responsible for buying GLV in their households (since in Nigeria this is more often women than men we decided to only include females), were adults (18 or older), were GLV consumers, and if they provided their informed consent. For the open market sample, in addition to the other inclusion criteria, respondents were included only when they indicated that they noticed Veg-on-Wheels vendors.

### Measures

For both the control and intervention periods, data were collected through a paper-based, self-administered questionnaire. The questionnaire for the control period included measures related to vegetable consumption behaviour and the determinants that impact this behaviour (known from the literature). During the intervention period, the focus mainly was on awareness, satisfaction with the Veg-on-Wheels intervention and the impact on consumption behaviour. While the questionnaires taken near the workplaces could often be administered in the offices, market questionnaires had to be taken on the street, the length of the questionnaire was reduced to better fit this context. **Table 1** provides an overview of the applied measures in both studies, with a detailed explanation provided in the following paragraphs.

**Control period: Vegetable consumption behaviour and its determinants.** *GLV purchase, consumption, and storage behaviourTo.* examine *vegetable-buying behaviour*, questions related to the following topics were included: (i) buying place for GLV; (ii) average travel distance to the most-visited GLV outlet in minutes/hours, including transportation mode; (iii) GLV buying frequency over the previous two weeks; and (iv) the number of GLV bundles that were bought over the previous two weeks.

To estimate the respondents' usual vegetable consumption behaviour, a Food Frequency Questionnaire (FFQ) was administered, similar to that of Van Assema and colleagues [35]. The FFQ measured the respondents' usual vegetable intake over the previous two weeks. To tailor the FFQ as much as possible to local perceptions and definitions of vegetables, vegetables were defined further into different categories: (i) GLV; (ii) tomatoes, onions and peppers; (iii) other cooked vegetables; and (iv) other raw vegetables. The respondents indicated their usual consumption frequency for each of these categories over the previous two weeks and the usual consumption amount in number of serving spoons. These data were converted in two steps to determine the intake level for each category: converting intake levels into meaningful data (portion sizes) and multiplying the intake frequency by portion sizes. In the third step, intake from the different categories was summed up.

Vegetable-storing behaviour for both fresh and cooked GLV was measured using five questions that covered the following topics: (i) whether the respondent stored fresh/cooked GLV;

**Table 1. Overview of the structure of the used questionnaires in the control and intervention period.**

| Topic | Measure | # items | Scale | Reliability |
|---|---|---|---|---|
| **Control period** | | | | |
| Self-reported behaviour | Buying behaviour | 8 | n.a. | n.a. |
| | Storage behaviour | 5 | n.a. | n.a. |
| | Consumption behaviour[a] | 8 | n.a. | n.a. |
| Socio-psychological determinants | Food Choice Motives[b] | 48 | 7-point Likert scale: from 1 = not important at all to 7 = extremely important | Between 0.70 and 0.89[1] |
| | Perceived barriers | 18 | 5-point Likert scale: 1 = never to 5 = always | n.a. |
| | Perceived product inconvenience[c] | 3 | 7-point Likert scale: from 1 = strongly disagree to 7 = strongly agree | 0.91 |
| | Self-efficacy[d] | 9 | 7-point Likert scale: from 1 = strongly disagree to 7 = strongly agree | 0.69 |
| Socio-demographics | Socio-demographics | 7 | n.a. | N.a. |
| **Intervention period[2]** | | | | |
| Awareness | Awareness intervention | 1 | Binary (yes/no) | n.a. |
| Self-reported behaviour | Consumption behaviour[2,a] | 8 | n.a. | n.a. |
| | Perceived consumption change[2] | 5 | Single item (decreased, did not change, increased) | |
| | Veg-on-Wheels buying | 1 | Binary (yes/no) | n.a. |
| Socio-psychological determinants | Attitude[e] | 6 | 7-point Likert scale: from 1 = strongly disagree to 7 = strongly agree | 0.97 (FUTA and secretariats), 0.85 (market) |
| | Barriers | 13 | 7-point Likert scale: 1 = strongly disagree to 7 = strongly agree | n.a. |
| | Motives | 11 | 7-point Likert scale: 1 = strongly disagree to 7 = strongly agree | n.a. |
| | Consumers' satisfaction | 2 | binary | n.a. |

[1] Health (0.888), Functional health (0.833), Mood (0.830), Convenience of preparation (0.880), Convenience of accessibility (0.757), Sensory appeal (0.863), Natural content (0.701), Price (0.811), Weight control (0.741), Familiarity (0.856), and Food safety (0.835).

[2] Two different versions of questionnaires were used in the intervention period, respondents at the open market filled out a shorter version of the questionnaire compared to FUTA and Secretariats since they are on the move and willing to spend less time. The measures Consumption behaviour and Perceived consumption change were therefore not included in the Open market questionnaire.

Based on questionnaires developed by

[a]Van Assema et al. (2002)

[b]Steptoe et al. (1995)

[c]Olsen et al. (2007)

[d]Raaijmakers et al. (2018), and

[e]Crites et al. (1994).

(ii) how long the respondent stored fresh/cooked GLV; and (iii) how the respondent stored fresh/cooked GLV.

*Determinants of GLV intake*: *motives, barriers, and self-efficacy*. *Food choice motives* (FCMs) concerning GLV were assessed using an adapted version of the original Food Choice Questionnaire (FCQ) [36]. Considering that previous qualitative research conducted in Nigeria indicated that the original FCQ might not fit the local context, the FCQ was extended based on these outcomes. The adapted FCQ comprised 48 items representing 11 dimensions, each of which was introduced with the affirmative sentence: 'It is important to me that the food I eat on a typical day. . .', followed by each motive. Examples of additional items are 'gives me energy' (functional health), 'is easy to wash/clean' (convenience of preparation), 'can be bought on markets, at road stalls and in shops close to where I live or work' (convenience of

accessibility) and 'is handled in a hygienic way' (food safety). Cronbach's alphas were sufficiently high for all dimensions showing a good inter-reliability of the scale (Table 1).

*Perceived barriers to GLV consumption in general* were measured with a scale that included 18 different possible barriers related to GLV and/or other vegetable consumption found in the literature, particularly previous conducted qualitative research in Nigeria. Each item was introduced by the question 'How often do you abandon buying GLV because. . .', followed by a potential barrier.

The *perceived inconvenience of GLV* was measured using three items based on the perceived product inconvenience scale [37], but adapted for GLV. An example of an item is 'It is difficult to plan, provide, prepare and cook GLV for a meal'. Exploratory factor analysis (EFA) revealed one factor, with a total explained variance of 84.9%.

The respondents' *self-efficacy*, i.e., their own ability to prepare and increase their vegetable consumption, was measured using a nine-item questionnaire [38]. Self-efficacy scales need to be tailored to the research topic. We used a scale developed earlier to measure self-efficacy in relation to vegetable intake in Nigeria [19]. An example of an item is 'A lot of vegetables are difficult to cook'. After recoding the negatively formulated items, EFA indicated a two-factor structure based on the Eigenvalue's scree plot, which explained 59.1% of the variance. After examining the results, it was decided to use one factor structure, as one factor included all the recoded items suggesting that is was a methodological- instead of content-based structure. The item 'I have a cook who prepared the vegetables for me' was deleted, as the Cronbach's α increased from 0.492 to 0.690.

**Intervention period: Veg-on-Wheels Intervention.** *Intervention outcomes*: *awareness and vegetable consumption. Awareness of the Veg-on-Wheels intervention* was measured with the question 'Did you see the "Veg-on-Wheels" vendors with their bikes and/or pushcarts (see picture) on the streets and in the surroundings?' using a binary scale (yes/no).

To estimate the respondents' usual *vegetable consumption behaviour* during the course of the Veg-on-Wheels intervention, the same FFQ as during the control period was administered. Next, buyers during the Veg-on-Wheels intervention (those that bought vegetables from Veg-on-Wheels) were asked whether they thought their buying and consumption behaviour with GLV had changed. The perceived consumption change was measured using five items. Each item was introduced with the affirmative sentence 'Compared with what I did before, due to the Veg-on-Wheels. . .', followed by each item (e.g., 'The number of times I ate GLV) and answered by decreased, did not change or increased. This question was in the questionnaire only at the FUTA and Secretariat sites.

*Determinants of intervention outcomes*: *attitude, barriers, and motives, satisfaction with intervention*. The *attitude* towards the Veg-on-Wheels intervention was measured using a six-item questionnaire–three items for cognitive attitude (interesting, useful and beneficial) and three items for affective attitude (attractive, good and favourable) [39]. The use of bipolar items (e.g., good/bad) was not well-understood when piloting the questionnaire; therefore, we used a seven-point answering scale instead. For the FUTA and Secretariat sites, the items loaded on a single factor and explained 86.5% of the variance, with all items contributing to the scale. Similarly, for the market data, the items loaded on a single factor and explained 57.7% of the variance, with all items contributing to the scale.

*Motives* for buying GLV from Veg-on-Wheels were measured among the buyers using a scale that included 11 different motives, which were found in the literature and qualitative research that the authors conducted previously. Each item was introduced by the sentence 'I bought vegetables from the Veg-on-Wheels vendor mainly because. . .', followed by a potential motive. Next, respondents were asked to suggest areas for improvement for Veg-on-Wheels.

*Barriers* to buying GLV from Veg-on-Wheels were measured among non-buyers using a 13-item scale that included different barriers related to GLV and/or other vegetable consumption. These obstacles were found in the literature and in qualitative research that the authors conducted previously. Each item was introduced with the sentence 'I didn't buy the vegetables from the Veg-on-Wheels vendor mainly because. . .', followed by a potential barrier. The answer category 'I have never seen Veg-on-Wheels' was added, and tips were asked on how to improve Veg-on-Wheels.

Consumers' *satisfaction* was measured using the following two items: 'Would you recommend the vegetables from Veg-on-Wheels to friends, family or colleagues?' and 'Would you continue buying vegetables from Veg-on-Wheels?'

**Data analysis.**   During the control period, data were collected from the FUTA and Secretariat sites. Control period data were used for descriptive analyses on GLV buying and storage behaviour. During the intervention period, data on intervention awareness and perception were collected from the FUTA and Secretariat sites, and the open market. These were described, and the differences between buyers and non-buyers were analysed for the locations separately using t-tests in SPSS 25 software. At the market location, demographic differences between buyers and non-buyers were tested using Chi-square. Intake data were collected only at the FUTA and Secretariat sites and were compared with t-test between buyers and non-buyers, and before and after intervention.

## Results

### Control period: Vegetable consumption behaviour and its determinants

Below, the results are presented of the baseline study providing insights in vegetable buying and storage behaviour. These descriptive results provide insights in buying location, transportation way and time, home grown of GLV, storage, average intake. Socio-psychological determinants of intake are also reported: reasons for abandoning GLV purchase, food choice motives, GLV inconvenience, and self-efficacy.

**Study sample.**   Altogether, 680 female respondents were included during the study's control period. As shown in the demographics on Table 2, 62.1% were married, 56.7% were highly educated, almost a quarter were students and on average, 4.4 people lived in their households.

**Vegetable buying, consumption and storage behaviour.**   All respondents indicated buying their GLV, while 52.4% of them also indicated that they grow their own. The frequency of buying GLV was most commonly one, two or three times per week, with 2.3 times per week on average (SD = 1.6). When GLV were purchased, on average, about 2.7 bundles of these vegetables were bought (SD = 1.9). More than 70% of the respondents indicated buying their GLV at one outlet. The open market was mentioned as the most common place, followed by street vendors and neighbourhood markets. Wholesale open markets, chain supermarkets, mini-supermarkets and other places, such as farmers supplies, were mentioned less often as places to buy GLV. Most respondents took a taxi or bus to their GLV outlet, or they walked. A minority of respondents drove there by car or motorcycle. Finally, 11 respondents indicated that they bought their GLV mostly through a home delivery service. Travel time was 30 minutes or less for 94.3% of the respondents, and 53.1% travelled 10 minutes or less (Table 3).

About one-third (32.1%) of the respondents said they stored their vegetables while the others ate them the same day. Fresh vegetables were stored on average from 2.3 days (SD = 1.3) and cooked GLV were stored on average for 2.5 days (SD = 1.4). When stored, the freezer was the most used method of GLV storage, followed by keeping them at room temperature or in the refrigerator. Keeping them outside was the least-often-applied method.

**Table 2. Overview socio-demographics study sample of the control study (percentages).**

| | | FUTA and Secretariats % of sample (N = 680) |
|---|---|---|
| **Family status** | Married | 62.1 |
| | Single | 36.2 |
| | Widow | 1.8 |
| **Number of people in the household** | Mean (sd); range 1–10 | 4.4 (2.1) |
| **Children <18 in the household** | Yes | 63.7 |
| | No | 36.3 |
| **Educational level[1]** | None | 2.8 |
| | Primary school | 2.4 |
| | Secondary school | 25.1 |
| | Polytechnic: OND | 13.1 |
| | Polytechnic: HND | 11.8 |
| | University (BSc) | 37.8 |
| | Post-university (MSc or PhD) | 7.1 |
| **Occupation** | Senior management/admin | 6.2 |
| | Manager | 1.5 |
| | Professional | 18.4 |
| | Skilled worker | 14.3 |
| | Unskilled worker | 7.1 |
| | Clerical worker | 7.4 |
| | Unemployed | 6.2 |
| | Student | 24.1 |
| | Other | 15.0 |
| **Age** | 18–20 | 9.0 |
| | 21–25 | 22.6 |
| | 26–30 | 9.6 |
| | 31–35 | 11.3 |
| | 36–40 | 13.5 |
| | 41–50 | 19.6 |
| | 51–55 | 14.4 |

[1] Primary school and below was considered low educated, secondary school as middle, and OND and above as high educated.

Examining average consumption per week, most respondents ate GLV, on average, 2.7 times per week (SD = 1.7) and ate on average 2.8 serving spoons each time (SD = 1.4), resulting in an average GLV intake of 8.3 serving spoons per week (approximately 415 g). Respondents used the full range of answering scales, with scores ranging between 0 and 49 spoons per week. GLV intake was 25.3% of the total vegetable intake of 32.8 spoons per week. Respondents also reported eating, on average, 13.8 (SD = 10.6) spoons of tomatoes, onions and peppers; 6.1 (SD = 7.4) spoons of other heated vegetables; and 4.6 (SD = 6.5) spoons of raw vegetables per week.

When examining the reasons for abandoning GLV purchases, almost a quarter of the respondents indicated that they often or always stopped buying GLV because 'they look like they used fertiliser or insecticide on it' (**Fig 1**). Other reasons cited for often or always ceasing

**Table 3. Green leafy vegetable (GLV) buying and storage behaviour at FUTA and Secretariats during control period (percentages of sample n = 680).**

|  |  | % of sample |  |
|---|---|---|---|
| **Purchase behaviour** |  |  |  |
| **Frequency of buying GLV** | Never | 0.1 |  |
|  | One day in two weeks | 6.3 |  |
|  | One day per week | 30.6 |  |
|  | Two days per week | 24.7 |  |
|  | Three days per week | 21.0 |  |
|  | Four days per week | 8.1 |  |
|  | Five days per week | 3.5 |  |
|  | Six days per week | 1.6 |  |
|  | Every day | 3.7 |  |
|  | More than once a day | 0.3 |  |
| **Number of GLV bundles per purchase** | One | 17.6 |  |
|  | Two | 48.8 |  |
|  | Three | 12.9 |  |
|  | Four | 10.4 |  |
|  | Five or more | 10.0 |  |
| **Outlet (% yes; more than one allowed)** | Open market | 67.2 |  |
|  | Street vendors | 29.6 |  |
|  | Neighbourhood markets | 26.3 |  |
|  | Wholesale open markets | 12.4 |  |
|  | Chain super markets | 4.1 |  |
|  | Mini-supermarkets | 2.1 |  |
|  | Other (farmer supply...) | < 1 |  |
| **Transportation mode (%)** | Taxi or bus | 48.2 |  |
|  | Walking | 30.1 |  |
|  | Car | 12.9 |  |
|  | Motorcycle | 6.9 |  |
|  | home delivery service | 1.7 |  |
|  | Other (own garden) | < 1 |  |
| **Travel time (%)[1]** | None, 0 minutes | 11.3 |  |
|  | 1–5 minutes | 21.1 |  |
|  | 6–10 minutes | 23.8 |  |
|  | 11–20 minutes | 28.2 |  |
|  | 21–30 minutes | 13.2 |  |
|  | More than 30 minutes | 2.4 |  |
| **Storage behaviour** |  |  |  |
| **Storage time of fresh / cooked GLV[1]** |  | **Fresh** | **Cooked** |
|  | Eat the same day | 78.7 | 81.2 |
|  | One day | 3.5 | 4.0 |
|  | Two days | 12.1 | 9.1 |
|  | Three or more days | 5.7 | 5.9 |
| **Storage method for fresh / cooked GLV (%)** |  | **Fresh** | **Cooked** |
|  | Freezer | 7.9 | 9.4 |
|  | Refrigerator | 5.4 | 5.4 |
|  | Room temperature | 9.1 | 5.9 |
|  | Keeping outside | 3.2 | 0.9 |

[1]Travel time was reported in minutes and answers combined into categories; similar storage time was reported in days.

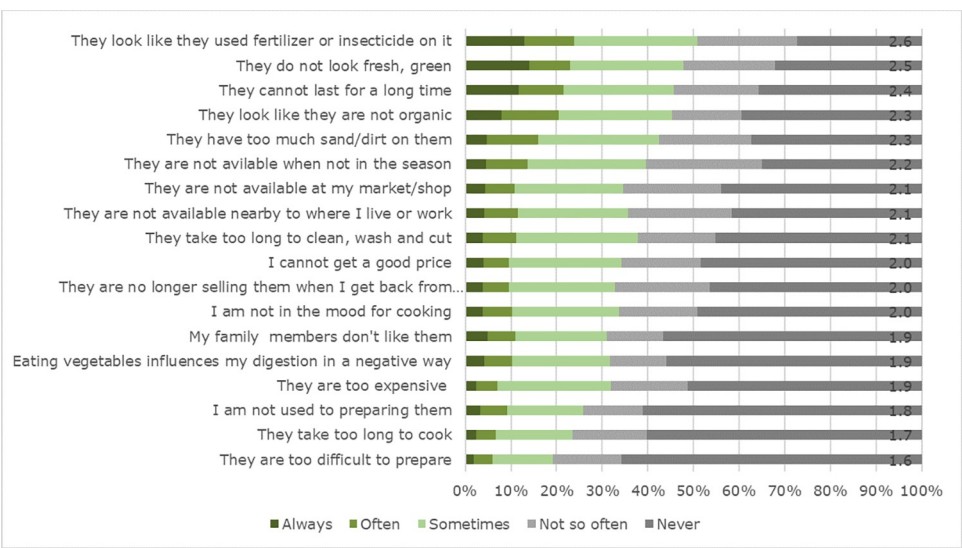

**Fig 1. Reasons for abandoning buying GLV (N = 680, control period) in percentages and mean values added at the end of the bar.**

GLV purchases were 'they do not look fresh, green' (23.5%), 'they cannot last for a long time' (22.8%), 'they look like they are not organic' (21.9%) and 'they have too much sand/dirt on them' (16.9%).

**Socio-psychological determinants.** Regarding FCMs, respondents overall viewed all motives as moderate to very important in their daily food choices. The motives food safety (M = 6.10, SD = 1.16), health (M = 6.06, SD = 1.03) and functional health (M = 6.02, SD = 1.01) were viewed as the most important, followed by convenience of accessibility (M = 5.89, SD = 1.24) and convenience of preparation (M = 5.86, SD = 1.19). Sensory appeal (M = 5.62, SD = 0.98), natural content (M = 5.54, SD = 1.10), weight control (M = 5.47, SD = 1.18), mood (M = 5.41, SD = 1.12), price (M = 5.30, SD = 1.16) and familiarity (M = 5.16, SD = 1.28) were viewed as somewhat less important motives. Respondents did not view GLV as inconvenient (M = 2.58, SD = 1.67) and indicated that they were confident that they had the knowledge, skills and ability to prepare vegetables (M = 5.08, SD = 1.03).

## The Veg-on-Wheels Intervention

Below the results are presented of the intervention outcomes. Vegetable consumption was compared between buyers from Veg-on-Wheels and non-buyers. Other measures described are: perceived change in intake (for buyers), intervention visibility, attitudes towards the intervention, and main motives and barriers for buying from the intervention.

**Vegetable consumption Veg-on-Wheels' buyers and non-buyers.** During the final intervention week, the average GLV intake was 9.4 serving spoons per week. Veg-on-Wheels customers ate, on average, 10.8 portions of GLV per week, which was significantly higher than non-buyers' intake, at 8.0 portions per week (Table 4). Buyers also ate a significantly wider variety of vegetables compared with non-buyers, including onions, peppers, tomatoes, other heated vegetables and raw vegetables. Around half the buyers perceived a positive effect from Veg-on-Wheels on their vegetable consumption: They agreed with the statements that compared with what they did before, due to Veg-on-Wheels, the number of times they ate GLV increased (57.6%), the portion sizes they ate increased (52.9%), the quality of the GLV that they ate increased (66.8%) and the healthiness of their food intake increased (73.2%). The

**Table 4. Mean values of self-reported vegetable consumption at FUTA and Secretariats during control and intervention period.**

| | | Control | Intervention | | | |
|---|---|---|---|---|---|---|
| | | Total (N = 680) | Total (N = 596) | Buyers (N = 295) | Non-buyers | t-value (buyers vs. non-buyers) |
| **Green leafy vegetables** | Mean | 8.3 | 9.4 | 10.8 | 8.0 | t = 4.2*** |
| | SD | 7.2 | 8.0 | 8.4 | 7.3 | |
| **Tomato, onion, pepper** | Mean | 13.8 | 15.7 | 17.5 | 14.0 | t = 3.6*** |
| | SD | 10.6 | 12.0 | 12.7 | 11.0 | |
| **Other heated vegetables** | Mean | 6.1 | 6.9 | 8.5 | 5.4 | t = 4.5*** |
| | SD | 7.4 | 8.7 | 9.8 | 7.1 | |
| **Raw vegetables** | Mean | 4.6 | 6.2 | 8.1 | 4.4 | t = 5.4*** |
| | SD | 6.5 | 8.6 | 10.0 | 6.6 | |
| **Vegetables total[1]** | Mean | 32.8 | 38.3 | 45.0 | 31.7 | |
| | SD | 21.5 | 27.8 | 30.6 | 23.1 | |

[1] Total vegetable intake is a sum score of the other categories over a period of the last week.

*** $p < 0.001$.

other half of the buyers perceived no changes in their GLV consumption behaviour, with one buyer believing that her intake had decreased, and two buyers thinking that their buying frequency had decreased. Concerning buying frequency from their regular seller, 16.3% indicated that this had decreased, 42.0% said that it had not changed and 41.7% reported an increase.

**Awareness and attitudes towards Veg-on-Wheels.** At the FUTA and Secretariat sites, Veg-on-Wheels visibility was high. Obviously, all buyers noticed the pushcarts and bikes, but 93.4% of the non-buyers (sample N = 596) also noticed them. Both buyers and non-buyers also had very positive attitudes towards 'Veg-on-Wheels (6.2 and 6.0 for buyers and non-buyers, respectively, on a scale from 1 to 7; SD = 0.6 and 1.2 respectively). Buyers' attitudes were significantly more positive than those of non-buyers (t = 2.5; $p = 0.01$). Buyers also showed high satisfaction with Veg-on-Wheels: 98.6% agreed with the statement, 'Would you recommend "Veg-on-Wheels" to friends, family or colleagues', and 97.6% answering that they would continue buying from Veg-on-Wheels.

At the markets, all respondents that were included indicated that they had seen Veg-on-Wheels vendors, as this was a criterion to participate in the study. Also at the market, buyers indicated high satisfaction with Veg-on-Wheels, with 96.9% (all but three respondents) agreeing with the statement 'Would you recommend "Veg-on-Wheels" to friends, family or colleagues' and said that they would continue buying from Veg-on-Wheels. Buyers (n = 96) and non-buyers (n = 1–8) at the market differed in age (t = 6.0; $p < 0.001$), educational level ($X^2 = 43.3$; $p < 0.001$), and current employment ($X^2 = 21.3$; $p = 0.003$). Compared to non-buyers, Veg-on-Wheels buyers were more often middle aged (especially in the 36–40 age category) and less often younger than 26 or older than 41. Also, they were more often higher educated (especially BSc or above) and less often had lower educational levels (especially none or primary school). Finally, they were more often managers or skilled workers and less often unskilled workers compared to other employment categories.

**Veg-on-Wheels buying barriers and motives.** Buyers at all the different locations (FUTA, Secretariat and market sites) strongly agreed with all motives to buy from Veg-on-Wheels related to quality, hygiene, appearance, health, curiosity to try, trust and convenience. Most items had average scores above 6 on a scale of 1 to 7, except items that 'could be bought on my way home', 'were convenient to cook' and 'were good value for money', which scored slightly below 6 on average (Table 5). Non-buyers overall (total sample) on average rated all

**Table 5. Motives and barriers for buying/not buying from 'Veg-on-Wheels' (mean values).**

| Motives for buying from the 'Veg-on-Wheels' | | Total (N = 391) | FUTA[1] and Secretariats[1] (N = 295) | Market[2] (N = 96) | T-value FUTA and Secretariats vs. Market |
|---|---|---|---|---|---|
| They seemed to have a high quality | Mean | 6.29 | 6.346 | 6.11[b] | T = 2.523* |
| | SD | 0.79 | 0.64 | 1.10 | |
| They seemed to have a good hygienic quality | Mean | 6.24 | 6.302 | 6.03[b] | T = 2.999** |
| | SD | 0.78 | 0.77 | 0.76 | |
| They looked green and fresh to me | Mean | 6.18 | 6.20 | 6.13[ab] | T = 0.681 |
| | SD | 0.94 | 0.96 | 0.86 | |
| I want to eat more fresh green leafy vegetables | Mean | 6.15 | 6.14 | 6.19 | T = -0.404 |
| | SD | 1.09 | 1.14 | 0.97 | |
| I wanted to try these vegetables | Mean | 6.14 | 6.29 | 5.70[b] | T = 5.383*** |
| | SD | 0.97 | 0.88 | 1.08 | |
| I want to eat more healthy | Mean | 6.12 | 6.10 | 6.16[ab] | T = -.443 |
| | SD | 1.05 | 1.09 | 0.92 | |
| I trusted the vendor | Mean | 6.07 | 6.20 | 5.65[b] | T = 4.209*** |
| | SD | 0.97 | 0.84 | 1.21 | |
| They were easy to buy | Mean | 6.06 | 6.108 | 5.91 | T = 1.818 |
| | SD | 0.95 | .9231 | 1.02 | |
| They could be bought on my way home | Mean | 5.99 | 6.071 | 5.76[b] | T = 2.561* |
| | SD | 1.04 | 1.023 | 1.06 | |
| They were convenient to cook | Mean | 5.99 | 5.969 | 6.05[a] | T = -.730 |
| | SD | 1.18 | 1.27 | 0.84 | |
| They were good value for money | Mean | 5.96 | 5.93 | 6.05[a] | T = -0.851 |
| | SD | 1.20 | 1.25 | 1.03 | |
| **Barriers for not buying from the 'Veg-on-Wheels'** | | Total (N = 389) | FUTA and Secretariats[1] (N = 281) | Market[2] (N = 108) | F(2,388) |
| I prefer to buy vegetables closer to my home | Mean | 3.81 | 3.53 | 4.55[a] | T = -5.048*** |
| | SD | 1.90 | 1.89 | 1.74 | |
| I never considered going | Mean | 3.75 | 3.49 | 4.43[a] | T = -4.529*** |
| | SD | 1.76 | 1.64 | 1.89 | |
| They were not good value for money | Mean | 3.54 | 3.35 | 4.05[a] | T = -3.257** |
| | SD | 1.91 | 1.84 | 1.98 | |
| I prefer to go to my own vendor | Mean | 3.52 | 2.97 | 5.01[a] | T = -11.240*** |
| | SD | 1.85 | 1.57 | 1.73 | |
| I prefer not to use pre-cut vegetables | Mean | 3.43 | 2.82 | 5.05[a] | T = -11.881*** |
| | SD | 1.82 | 1.45 | 1.73 | |
| I didn't know what they were selling | Mean | 3.38 | 3.35 | 3.47 | T = -0.647 |
| | SD | 1.69 | 1.63 | 1.83 | |
| I prefer not to use pre-washed vegetables | Mean | 3.25 | 2.88 | 4.23[a] | T = -6.561*** |
| | SD | 1.61 | 1.26 | 1.99 | |
| They seemed to have a low quality | Mean | 3.12 | 3.10 | 3.18[ab] | T = -0.416 |
| | SD | 1.46 | 1.35 | 1.72 | |
| I didn't trust the vendor | Mean | 3.02 | 2.27 | 4.99[a] | T = -13.547*** |
| | SD | 1.78 | 0.88 | 2.02 | |
| I didn't trust the sources | Mean | 2.86 | 1.99 | 5.28[a] | T = -20.236*** |
| | SD | 1.77 | 0.66 | 1.64 | |

(*Continued*)

**Table 5.** (Continued)

| Motives for buying from the 'Veg-on-Wheels' | | Total (N = 391) | FUTA[1] and Secretariats[1] (N = 295) | Market[2] (N = 96) | T-value FUTA and Secretariats vs. Market |
|---|---|---|---|---|---|
| They seemed to have a low hygienic quality | Mean | 2.81 | 2.64 | 3.26[a] | T = -3.242*** |
| | SD | 1.47 | 1.27 | 1.82 | |
| I eat enough vegetables | Mean | 2.60 | 2.09 | 4.03[a] | T = -9.172*** |
| | SD | 1.58 | 0.87 | 2.13 | |
| They didn't look green and fresh to me | Mean | 2.53 | 2.36 | 2.99[a] | T = -3.835*** |
| | SD | 1.17 | 0.89 | 1.63 | |

[1] In total N = 596 were interviewed at or in the surroundings of FUTA and Secretariats, from whom N = 295 bought from 'Veg-on Wheels', N = 281 did not, and N = 20 did not saw the 'Veg-on-Wheels'.

[2] In total N = 204 were interviewed Market, from whom N = 96 bought from 'Veg-on-Wheels', N = 108 did not and N = 19 did not saw the 'Veg-on-Wheels'.

Note: 7-point Likert scale from 1 = strongly disagree to 7 = strongly agree.

* p < 0.05

** p < 0.01

*** p < 0.001. [abc] different letters indicate da significant difference between study locations.

Note We looked also at the differences between FUTA and Secretariats and found marginal differences between these two locations.

barriers to not buying from Veg-on-Wheels below or slightly below the scale middle (4). In examining the barriers separately, respondents most often agreed with statements related to vendors and trust ('I did not trust the vendor', 'I did not trust the sources', 'I prefer to go to my own vendor') and that they did not like the pre-cut vegetables (**Figs 2 and 3**). Significant differences were found between the locations. More specifically, the most significant differences were found between the market site and the FUTA and Secretariat sites. Trust in Veg-on-Wheels vendors, the source of the GLV and a preference to go to one's own vendor were viewed as more important at the market site rather than at the FUTA and Secretariat sites (**Table 5**).

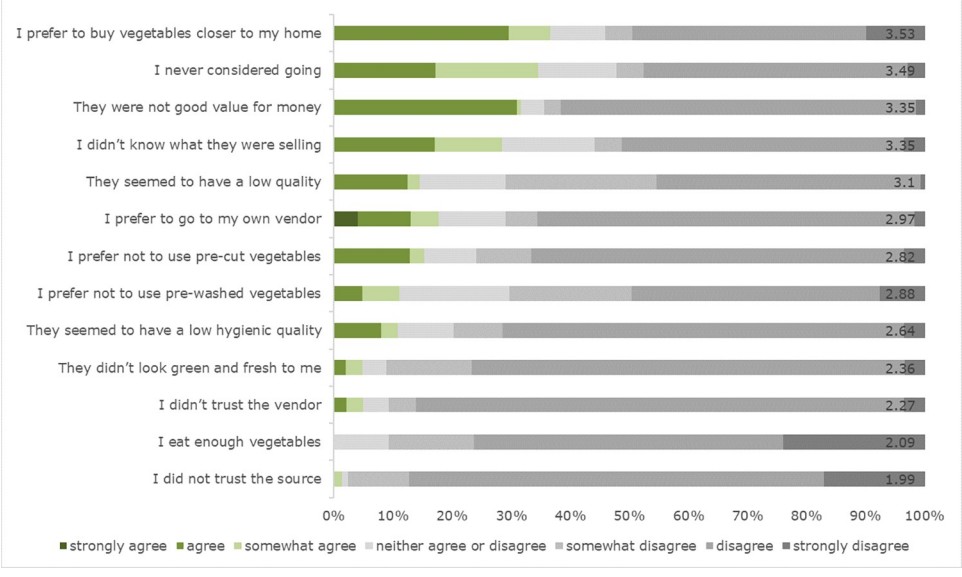

**Fig 2. Reasons for not buying GLV from the intervention near workplaces at FUTA university and (state) secretariats (N = 295) in percentages and mean values added at the end of the bar.**

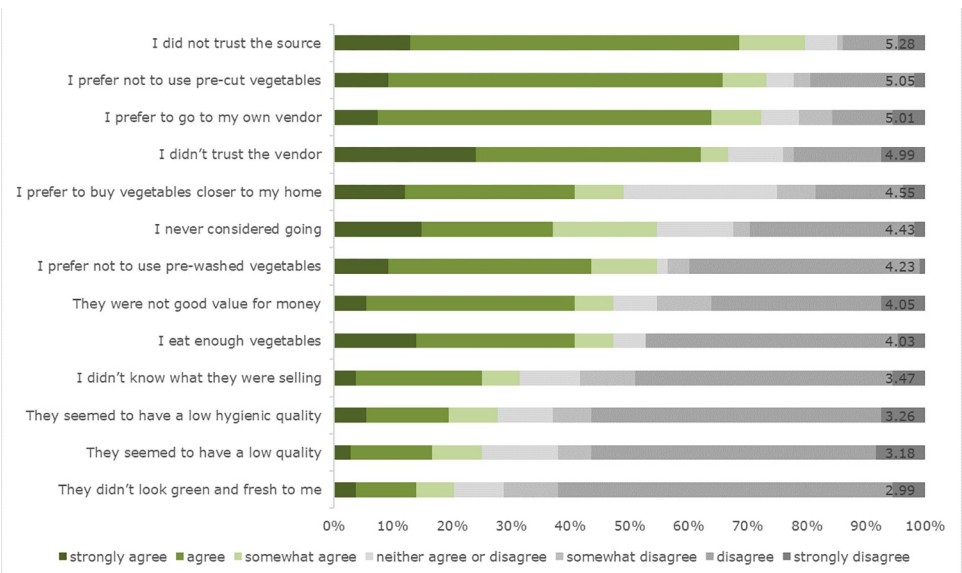

**Fig 3. Reasons for not buying GLV from the intervention at the market (N = 108) in percentages and mean values added at the end of the bar.**

Several tips from participants for the Veg-on-wheels involved better sales, including marketing, publicity, branding, different marketplaces and persuasive sellers. Others mentioned issues that might function as barriers, such as the need for availability, freshness at all times, removal of yellow leaves, information on where the vegetables are coming from, contact information, packaged uncut vegetables, selling other necessities (e.g., tomatoes and peppers) and selling vegetables without fertiliser.

## Discussion

### Summary of main intervention outcomes and implications

This explorative intervention study showed that the Veg-on-Wheels intervention was successful in selling cooled, pre-cut and washed GLV to urban Nigerians and that buyers consumed significantly more GLV than non-buyers. Most buyers also perceived that their intake had increased due to the intervention. The intervention had a high visibility, and both buyers and non-buyers had positive attitudes. These results are in line with studies from other countries suggesting the importance of the food environment in shaping consumer behaviour and showing that interventions at the workplace environment can increase availability and are promising strategies to increase consumption of healthy food products, such as vegetables [9,13,14]. Next, it was found that intake of other vegetables–such as tomatoes, onions and peppers–also was significantly higher among the buyers compared with non-buyers. This finding seems to be logical, as GLV are likely to be consumed as part of traditional vegetable soups and/or stews that include these ingredients [40]. The study also provided general insights in vegetable buying and handling behaviour. Most consumers bought from the open market, followed by street vendors and neighbourhood markets. Also half of them grew their own. The majority took a taxi or bus to this outlet or walked and usually this took less than 30 minutes, for half of them less than 10 minutes. Storage was not uncommon, but usually vegetables were eaten the same day. Average GLV intake was 8.3 serving spoons per week (approximately 415 g) with large differences between respondents. The main reasons for abandoning GLV purchase was use of pesticides and freshness.

## Health motives and convenience barriers

The Veg-on-Wheels design was based on prior research that demonstrated the importance of health as a motive, and convenience and availability as barriers to vegetable consumption [18,19,20]. The control period provided us with more detailed insights into urban Nigerians' vegetable buying and consumption behaviour, which helps in better understanding the mechanisms behind the intervention outcomes. Although GLV are common in Nigerian dishes, these vegetables were bought 2.3 times per week on average and most-often eaten the same day, suggesting that there is room to increase purchase frequency, which would result in higher intake frequency. In an earlier study, when distance to the market decreases, consumption of GLV increases significantly [15]. Indeed, convenience was viewed as an important motive in food choices, but at the same time, travel time to the open market was less than 10 minutes for half the respondents and less than half an hour for most. Furthermore, respondents did not think that GLV were inconvenient to acquire and prepare. These results clash with those from previous studies showing that GLV preparation is viewed as time-consuming [41]. Rather than access and preparation, it could be an issue of freshness and shelf life, considering that this was mentioned as a principal barrier to not buying GLV at baseline. In line with this finding, Veg-on-Wheels was successful at selling not only at the FUTA and Secretariat sites, which were near workplaces and far from open/traditional markets, but also at the open market. Furthermore, people who made the effort in time and transportation costs to come to the market were interested in the cooled and pre-cut vegetables. It also seemed that other motives also were driving purchases, mostly related to health and food safety (adulteration), as the use of pesticides and not being organic were mentioned as frequent barriers to GLV purchases. The main motives for buying were quality, hygiene, appearance, health, curiosity to try, trust and convenience while the main barriers were related to trust.

## The issue of trust

Trust in the vendor, GLV source and consumers' preferences to patronise their own vendors were important motives for buyers and non-buyers of Veg-on-Wheels–and differed significantly depending on vending location. In our study, the intervention was branded as a FUTA intervention, so it is not surprising that trust was higher for consumers on the FUTA campus who work there on a daily basis, compared with the diverse population of respondents at the market. Trust was very important, and the intervention's innovation depends on the seller's trustworthiness. This result also is confirmed in the literature; consumers, regarding food safety issues, have been found to try to develop personal relationships with their food vendors to help ensure quality [18]. Furthermore, the fact that trust in the vendor plays an important role in the acceptance and purchase of GLV can be confirmed by prior studies that indicate a credible source, like a vendor, may be particularly persuasive with consumers during their consideration phase on whether to buy a product. Consumers then generally have not yet formed their opinions about the purchase, and a credible salesperson or vendor can help influence consumers' buying decisions [42,43].

## Study strengths, limitations and future research suggestions

This study has some strengths and limitations. One of its strengths is that it measures the effect from the Veg-on-Wheels intervention, but also can explain the intervention's success. Data on consumers' purchase, preference and consumption behaviour with GLV also were collected, which helped in better understanding the intervention's mechanisms. As a result, we know that other motives besides health, convenience and availability–namely food safety and trust–are more likely to drive purchases from and acceptance of Veg-on-Wheels. Future studies

could further explore the aspects that define consumer trust in the washed and pre-cut GLV. In our case it was the FUTA branding but other trustful sources could be similarly successful in selling the GLV. One of the limitations of this study is the use of self-reported measures of GLV and total vegetable intake. Although we chose to work with self-reported measures for practical reasons, this method has some disadvantages. For example, respondents may have forgotten details about their intake, and various biases may have affected the results, e.g., social desirability bias, or the possibility that some respondents guessed the study's aim and provided biased responses. Furthermore, in the Nigerian context, scanner data were not available, but future studies could use methods for intake recall over shorter time periods, such as diaries or 24-hour recall. These methods are more intensive and require follow-up from respondents, so it would not be feasible for large samples, like that of our study, but can be used in a smaller and more focussed study that aims to duplicate the present study's results. Another limitation in this study was that we were unable to perform within-person comparisons (within-subjects design). Practical issues hindered the identification of respondents during follow-up. For example, self-reported birth dates often are unreliable in Nigeria, and phone numbers and names could not be used to identify respondents because they shared offices and, therefore, phone numbers with other workers. Also, some participants moved, changed phone numbers or were out of town due to public holidays during the course of the study. Considering that the research setting did not enable a within-subjects design, a between-subjects analysis was used. A disadvantage of this is that we could not be sure whether the higher intake of vegetables among buyers compared with non-buyers actually was due to the Veg-on-Wheels intervention. It could be that these respondents who eat more vegetable were more likely to buy from the intervention. Therefore, future research is needed to replicate the results. Randomised controlled studies are difficult to conduct in this research setting, but it is possible to include a control group in a similar context with a sample of respondents with characteristics similar to those of the study population.

Finally, the sample's representativeness needs to be considered. It was interesting that Veg-on-Wheels buyers at the market were more likely to be highly educated and non-buyers were more likely to be less-educated. In examining the FUTA and Secretariat sites more specifically, buyers at these locations seem to be more convenience-oriented. Research conducted in urban Kenya also found that formal, full-time employed and/or businesspeople are more likely to consume less indigenous vegetables compared with unemployed people or casual labourers [44]. In our sample this also seemed to be the target group that was most interested in buying from Veg-on-Wheels. Furthermore, when a household's cook is formally employed outside the home, it is found that this negatively influences consumption intensity of GLV [15]. However, consumers have a more positive attitude toward GLV when they perceive them as nutritious [45]. Our study population at the FUTA and Secretariat sites and the consumer population at the market seem to be a non-average sample in that they were more highly educated and, on average, probably more motivated to pursue the health benefits from higher vegetable consumption. Therefore, replication of the results in other populations, including insights on their motives and barriers to GLV consumption, could be an interesting next step.

## Study implications and future directions from a food systems perspective

Being able to address nutrition challenges effectively, intervention strategies on healthy eating, or on vegetable consumption more specifically should be considered within a food systems perspective and not in isolation. Dietary behaviour related to consumer purchase behaviour is shaped in the context of the food environment and is based on the outcomes of these activities and drivers on a range of domains, such as food security, socioeconomics and the

environment/climate. Findings of a study conducted in Brazil suggest that new possibilities of interventions should combine individual and environmental strategies [9]. The Veg-on-Wheels intervention combined different components within the food system by intervening in the food environment (vegetable availability and physical access, and providing health information on GLV), combined with consumer behaviour (identifying consumers' motives, barriers and needs regarding vegetable consumption) and socio-cultural and socio-demographic drivers. This resulted in a successful intervention that has the potential to increase vegetable consumption among urban Nigerians living in Akure and provide a source of income and employment generation for the urban poor. Earlier studies have shown how selling food items has generated revenue for Nigerian urban dwellers [46,47]. For sustainable intervention in the long term, other elements of the food system should be included, such as food supply chains and political actions. GLV producers, mostly women from farming cooperatives, can be encouraged to increase production through improved knowledge of GLV production, irrigation and storage facilities. Organic and other natural vegetables are very important to buyers and are linked to trust. Next, vendors should be engaged to sell their products in a wider array of locations and/or use refrigerating facilities to keep vegetables fresh throughout the day. The appearance of GLV is viewed as an important motivation for buying GLV. Insights on the consumer perspective were useful not only in the design of the innovation, but also in understanding consumers' motives and barriers to buying from Veg-on-Wheels. Trust in the vendor played a major role in acceptance of GLV, particularly at the FUTA and Secretariat sites. As in the case of FUTA site, a venture with reputable institutions can be established to facilitate continuity.

## Conclusions

Current vegetable consumption in Nigeria is below recommendations, and this study showed that an innovation in the food environment to overcome consumers' motives, barriers and needs has the potential to increase vegetable consumption among urban Nigerians living in Akure. By intervening on availability, convenience and healthiness, the intervention targeted different consumer behaviour motives within the food environment as an entry point. The results show that a market exists for convenience vegetables and that they have the potential to increase vegetable intake. Interestingly, the intervention showed that trust in the vendor and food safety also were main drivers of purchases from Veg-on-Wheels. This study also provided insight into GLV and other vegetable consumption among urban Nigerians living in Akure, as it identified convenience, freshness, shelf life, health, food safety and trust in vendors and other sources as main motives and drivers for vegetable buying and consumption behaviour.

## Supporting information

**S1 File.**
(DOCX)

## Acknowledgments

We thank Prof T.N Fagbemi and Prof. I.B Oluwalana for their input during the conceptualization and their support with resources needed. We also thank our field team: Adeola Adeoye, Agunbiade Richard, Akinnagbe Seyi, Aransiola Peace, Buraimoh Samuel, Nwachukwu Uche-chi, Owolabi Adenike, Oyinloye Anuoluwapo, Temiloluwa Sanmi-Lawal, Teniola Temitope, Adelugba Victor, Ajao Ifeoluwa, Anjorin Niyi, Oyindamola Agun. Our Vendors: Solomon Ikhazobor, Gbohunmi Liasu-Oni, Oreoluwa Sanmi-Lawal, Simeon Fawoye, Courage Olamoju,

The Processors: Hakeem Gboyega, Jumoke Lawal and Ruth Orhoe, Kehinde Akande. Our data Entry Officer: Timileyin Akinjuyitan

## Author Contributions

**Conceptualization:** Harriette M. Snoek, Ireen Raaijmakers, Oluranti M. Lawal.

**Data curation:** Harriette M. Snoek, Ireen Raaijmakers, Oluranti M. Lawal.

**Formal analysis:** Harriette M. Snoek, Ireen Raaijmakers.

**Investigation:** Harriette M. Snoek, Ireen Raaijmakers, Oluranti M. Lawal.

**Methodology:** Harriette M. Snoek, Ireen Raaijmakers, Oluranti M. Lawal, Machiel J. Reinders.

**Project administration:** Harriette M. Snoek, Ireen Raaijmakers, Oluranti M. Lawal.

**Resources:** Oluranti M. Lawal.

**Supervision:** Harriette M. Snoek, Ireen Raaijmakers, Oluranti M. Lawal.

**Validation:** Machiel J. Reinders.

**Visualization:** Harriette M. Snoek, Ireen Raaijmakers, Machiel J. Reinders.

**Writing – original draft:** Harriette M. Snoek, Ireen Raaijmakers, Machiel J. Reinders.

**Writing – review & editing:** Harriette M. Snoek, Ireen Raaijmakers, Oluranti M. Lawal, Machiel J. Reinders.

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
