## [Decision Letter · Decision Letter 0]

6 Dec 2021

PONE-D-21-32704A proof-of-principle study with convenience vegetable in urban Nigeria - The Veg-on-Wheels interventionPLOS ONE

Dear Dr. Snoek,

Thank you for submitting your manuscript to PLOS ONE. After careful consideration, we feel that it has merit but does not fully meet PLOS ONE’s publication criteria as it currently stands. Therefore, we invite you to submit a revised version of the manuscript that addresses the points raised during the review process.

The manuscript is generally well written and suitable for PLOS One. However, several issues are to be addressed, e.g., introduction and results. Please revise the manuscript by including the reviewers' comments, and, if not possible, please explain why.

We look forward to receiving your revised manuscript.

Kind regards,

Laurentiu Rozylowicz, Ph.D.

Academic Editor

PLOS ONE

Journal Requirements:

Reviewers' comments:

Reviewer's Responses to Questions

**Comments to the Author**

1. Is the manuscript technically sound, and do the data support the conclusions?

Reviewer #1: Partly

Reviewer #2: Yes

2. Has the statistical analysis been performed appropriately and rigorously? 

Reviewer #1: I Don't Know

Reviewer #2: Yes

3. Have the authors made all data underlying the findings in their manuscript fully available?

Reviewer #1: No

Reviewer #2: No

4. Is the manuscript presented in an intelligible fashion and written in standard English?

Reviewer #1: No

Reviewer #2: Yes

5. Review Comments to the Author

Reviewer #1: The authors present an innovative study with interesting findings. The structure and detail of the manuscript need work, particularly with the reporting of methods and findings, and discussion of the results.

Abstract: The percentages of respondents in the post-intervention survey are unclear. Also, what parameters the surveys used to asses the success and other aspects of the intervention are unclear, as the abstract reports only the significant differences.

Introduction: This section is rather fragmented overall and does not flow cohesively towards the rationale of the intervention and the associated study. Specific suggestions:

Line 35: Knowledge and awareness among whom - the scientific community, the general public? Suggest rewording to distinguish scientific literature from public awareness.

Line 39: There must be more recent evidence from within a decade? As the next sentence contradicts this one, I suggest either updating the reference or deleting this sentence.

Line 43: Unless there is evidence to compare dietary diversity or frequency of vegetable consumption, this sentence is out of place with the rest of the paragraph - in fact, the entire passage does little to support the claim that vegetable consumption is low. Suggest either drawing out evidence to substantiate this, or reframing the passage to discuss the importance of vegetables and the lack of information on consumption.

Line 54: Suggest sticking to a food environments framing, as the study does not engage with the larger food systems approach. It might help to elaborate on the definition of food environments (and exclude food systems), and the various examples that are mentioned fleetingly without apparent linkages, e.g. line 84.

Line 89: Business service innovation is not exactly explained in the passage that follows, so suggest either adding content about the same, or merging sections 1.2, 1.3, and 1.4, as they essentially describe the food environments approach and the Nigerian context.

Line 108: How was the effectiveness measured - qualitatively or quantitatively?

Methods: Despite the level of detail in this section, there remain major gaps in description of the methods. What were the control and intervention samples? Were the workplace points of sale operated through the day, or only after hours? What were the recruitment criteria? Why were the data collection at the open market and workplace points of sale different? What descriptive and inferential analyses were undertaken, to answer what questions or test what hypotheses, using what software? The parameters measured (e.g. buying, consumption, storage behaviours; motives, barriers, perceptions, efficacy; awareness, attitude, satisfaction) need to be listed and justified either in the introduction or the methods before the detailed passages on the specific variables measured. This will provide further clarity on what the study was aiming to achieve, and how. It is good practice to append a survey questionnaire to help the reader understand how these variables were measured and questions framed to elicit responses. Minor suggestions:

Line 133: Was the pamphlet distributed independent of the vegetable sale?

Line 138: There were questionnaires and no interviews, so the correct word to use would be 'trained interpreter'

Line 142: Enhance awareness of?

Line 198: Either number or merge this section

Results:

Table 2: Mention in the heading and the column that the numbers presented are percentages.

Section 3.1.2: This description leaves the reader with several questions, e.g. How many responses were collected at each of the sites? What was the mean and sd? What was indeed the most common frequency of GLV purchase - one, two, or three times a week? What about the remainder 30% of respondents who did not buy GLVs at one outlet? Suggest condensing all this information into one or two tables or figures to make things more transparent. Some interesting findings here about drivers and barriers in GLV consumption.

Line 358: Are there numbers for these?

The discussion does well to include limitations, and the implications section could be strengthened once the methods and results are clearer, with greater reference to evidence on food environments (not systems) from the region and around the world.

Reviewer #2: The manuscript titled A proof-of-principle study with convenience vegetable in urban Nigeria - The Veg-on- Wheels intervention aims at presenting the results of a proof-of-principle study on the purchase of amaranth and fluted pumpkin leaves.

The study focuses on Nigeria, a country where vegetable intake is low due to various constrains ranging from affordability to changes in consumption patterns doe to cultural influences. The study documents well the situation in Nigeria and proposes a sound methodology.

What I found lacking is a clear identification of the research gap to be filled with this proof-of-principle study. To my knowledge, proof-of-principle studies aim at verifying that some concept/ theory/ intervention has practical potential. Authors need to stress more how their assessment complements existing knowledge in Nigeria and elsewhere.

The study procedure is clear. Authors proposed a quasi-experimental design with control and intervention

Periods. Respondents were recruited in the vicinity of the intervention setting sites, near workplaces. For the control period the questionnaire was adapted to match the Nigerian context considered that the food choice motives might differed than what was proposed in previous studies.

The results are presented in detail both in the text and through figures. While the section is organized by topics, due to the amounts of information the reader might find it difficult to pin point the take home message. Authors could consider writing a small paragraph at the beginning of the results with the main findings.

Small observations

Line 68 – what kind of design innovations?

Line 91 – what do you mean here by food environment?

Line 104 – Not sure it is a good idea to already draw the conclusions (i.e. Veg-on-Wheels contributed to increased vegetable intake) in the headlines of the introduction

Line 198 – please ad a few lines to explain the purpose of Cronbach`s alphas in this context.

Line 264 – authors need to explain why the sample contained women only. Please do so in the methods section.

Line 357 - see Error! Reference source not found.

6. PLOS authors have the option to publish the peer review history of their article (what does this mean?). If published, this will include your full peer review and any attached files.

Reviewer #1: **Yes: **Mallika Sardeshpande

Reviewer #2: **Yes: **Simona R Gradinaru

---

## [Author Response · Author response to Decision Letter 0]

5 Apr 2022

Responses to the Reviewers’ comments (see also cover letter document)

Please note that references to line numbers are for the document with track changes in “all markup” view. 

Reviewer #1: The authors present an innovative study with interesting findings. The structure and detail of the manuscript need work, particularly with the reporting of methods and findings, and discussion of the results. 

Abstract:

The percentages of respondents in the post-intervention survey are unclear. We adjusted the sentence and the presentation of the percentages of the respondents. See lines 21-24. 

Also, what parameters the surveys used to asses the success and other aspects of the intervention are unclear, as the abstract reports only the significant differences. About presenting only the significant differences and parameters in abstract, it was not completely clear what non-significant differences could be added: 

• For the motives and barriers reported, it was perhaps not clear if the aspects mentioned were motives or barriers (e.g. trust). We adjusted the text to make more clear what the barriers were. Not all barriers were mentioned, only the main ones and those that different between open market and work locations. See lines 27-30.

• Vegetables intake was compared between buyers and non-buyers of the intervention. Only green leafy vegetable intake was reported in the abstract but the other vegetables intake measures were also higher in the intervention group. This was now added. See lines 26-27.

Apart from that, we believe we included all important outcome measures. The intervention measures awareness and attitude were reported in the abstract. For reasons of space, perceived consumption change and consumer satisfaction were not included in the abstract but both were in favor of the intervention: respondents reported to eat more vegetables due to the intervention and were satisfied with it. 

Introduction: This section is rather fragmented overall and does not flow cohesively towards the rationale of the intervention and the associated study. Specific suggestions:

Line 35: Knowledge and awareness among whom - the scientific community, the general public? Suggest rewording to distinguish scientific literature from public awareness. Good point, this was not clear from the text and has been adjusted to “scientific knowledge”. See line 42.

Line 39: There must be more recent evidence from within a decade? As the next sentence contradicts this one, I suggest either updating the reference or deleting this sentence.

Line 43: Unless there is evidence to compare dietary diversity or frequency of vegetable consumption, this sentence is out of place with the rest of the paragraph - in fact, the entire passage does little to support the claim that vegetable consumption is low. Suggest either drawing out evidence to substantiate this, or reframing the passage to discuss the importance of vegetables and the lack of information on consumption. Indeed, we should add that recent data is lacking. In general there is a lack of reliable individual level food intake data in low and middle income countries. In Nigeria, national food surveys have not been conducted recently and therefore we included the reference of 2004 from the latest one. And indeed, as the reviewer points out we cannot be sure that intake is low in Nigeria at the time of the study due to the lack of data. We adjusted the text and rephrased the paragraph to make this more clear. We also included a more recent but more general reference on the intake of vegetables in sub-Saharan Africa (Mensah et al., 2021)1. See lines 45-59.

Line 54: Suggest sticking to a food environments framing, as the study does not engage with the larger food systems approach. It might help to elaborate on the definition of food environments (and exclude food systems), and the various examples that are mentioned fleetingly without apparent linkages, e.g. line 84.

Line 89: Business service innovation is not exactly explained in the passage that follows, so suggest either adding content about the same, or merging sections 1.2, 1.3, and 1.4, as they essentially describe the food environments approach and the Nigerian context. We agree that the food environment should be more prominent as the place when consumers interact with the food system, and since it was the entry point of our intervention. This part of the introduction was adjusted by shortening the part of food systems and deleted it from the title of the paragraph, and by adding literature on the food environment2,3,4,5,6. Paragraphs 1.2 - 1.4 were combined as suggested into two sections: “the food environment as an entry point for interventions” and “consumer behaviour in interactions with the food environment”. We also added a recent study by Blake et al., (2021) on the role of food choice in food system research7. We also deleted the part on “business services” since indeed it was confusing. The relevant information on mobile produce markets and interventions at workplaces was included in the part on the food environment. See lines 70-182. 

Line 108: How was the effectiveness measured - qualitatively or quantitatively? The effectiveness of veg on wheels to facilitate increased vegetable consumption was quantitative through survey. We added that to lines 193-194.

1Mensah, D. O., Nunes, A. R., Bockarie, T., Lillywhite, R., & Oyebode, O. (2021). Meat, fruit, and vegetable consumption in sub-Saharan Africa: a systematic review and meta-regression analysis. Nutrition reviews, 79(6), 651-692.

2 Westbury, S., Ghosh, I., Jones, H. M., Mensah, D., Samuel, F., Irache, A., ... & Oyebode, O. (2021). The influence of the urban food environment on diet, nutrition and health outcomes in low-income and middle-income countries: a systematic review. BMJ global health, 6(10), e006358.

3Turner, C., Aggarwal, A., Walls, H., Herforth, A., Drewnowski, A., Coates, J., ... & Kadiyala, S. (2018). Concepts and critical perspectives for food environment research: A global framework with implications for action in low-and middle-income countries. Global food security, 18, 93-101.

4Turner, C., Kalamatianou, S., Drewnowski, A., Kulkarni, B., Kinra, S., & Kadiyala, S. (2020). Food environment research in low-and middle-income countries: a systematic scoping review. Advances in Nutrition, 11(2), 387-397.

5 Curioni, C. C., Boclin, K. L. S., Silveira, I. H., Canella, D. S., Castro, I. R. R., Bezerra, F. F., ... & Faerstein, E. (2020). Neighborhood food environment and consumption of fruit and leafy vegetables: Pro-Saude Study, Brazil. Public health, 182, 7-12. 

6Downs, S. M., Ahmed, S., Fanzo, J., & Herforth, A. (2020). Food environment typology: advancing an expanded definition, framework, and methodological approach for improved characterization of wild, cultivated, and built food environments toward sustainable diets. Foods, 9(4), 532. Food and Agriculture Organisation of the United Nations – FAO (2016) Influencing food environments for healthy diets, Rome, Available at: http:// www.fao.org/3/a-i6484e.pdf 

7Blake, C. E., Frongillo, E. A., Warren, A. M., Constantinides, S. V., Rampalli, K. K., & Bhandari, S. (2021). Elaborating the science of food choice for rapidly changing food systems in low-and middle-income countries. Global Food Security, 28, 100503.

Methods

Despite the level of detail in this section, there remain major gaps in description of the methods. Thank you for the detailed questions, these were very helpful to fill the gaps in the methodology. Below we added line number of where the information is presented. Also, since several questions were about sample we put the information on respondents selection is a separate section starting on line 238.

What were the control and intervention samples? See paragraph on respondents selection starting on line 238.

Were the workplace points of sale operated through the day, or only after hours? Lines 216-217 “... and sold during office working hours (roughly from 9 to 5; including the end of the working day when there is less availability in open markets).”

What were the recruitment criteria? See paragraph on respondents selection. Lines 249-253. 

Why were the data collection at the open market and workplace points of sale different? While the questionnaires taken near the workplaces could often be administered in the offices, market questionnaires had to be taken on the street, the length of the questionnaire was reduced to better fit this context. This was added on lines 265-267. 

What descriptive and inferential analyses were undertaken, to answer what questions or test what hypotheses, using what software? Added on lines 370-371. 

The parameters measured (e.g. buying, consumption, storage behaviours; motives, barriers, perceptions, efficacy; awareness, attitude, satisfaction) need to be listed and justified either in the introduction or the methods before the detailed passages on the specific variables measured. This will provide further clarity on what the study was aiming to achieve, and how. The information is (added to) the methods on the following lines: buying [275-278], consumption [279-291 and 328-335], storage behaviours [275-278]; motives, [295-304 and 345-349] barriers [305-308 and 350-355], perceptions [309-312], efficacy [313-321]; awareness [325-327], attitude [337-344], satisfaction [361-363]. Also, in the introduction we added a reference to support self-efficacy8 since this was indeed lacking (lines 165-167) and made a more explicit link to the measures (lines 190-196). In introduction and methods we used “determinants” more consistently to make the link clearer between the determinants described in the introduction, measures in the methods, and results. 

It is good practice to append a survey questionnaire to help the reader understand how these variables were measured and questions framed to elicit responses. Several items were based on existing scales that have been published by the developers of the scales. We added these references to Table 1. 

8Michie, S., Van Stralen, M. M., & West, R. (2011). The behaviour change wheel: a new method for characterising and designing behaviour change interventions. Implementation science, 6(1), 1-12.

Minor suggestions:

Line 133: Was the pamphlet distributed independent of the vegetable sale? “...distributed around the selling locations by project team members who were not selling the vegetables and were attached to the bicycles and pushcarts”. See lines 222-223.

Line 138: There were questionnaires and no interviews, so the correct word to use would be 'trained interpreter' -> changed throughout the MS. 

Line 142: Enhance awareness of? -> awareness of their affiliation, this was added to the MS. Awareness of their affiliation is important to avoid that people might get suspicious or hostile towards the field team. See lines 231.

Line 198: Either number or merge this section -> indeed we forgot to number for this caption and also it could be deleted and merged with the previous section. 

Results:

Table 2: Mention in the heading and the column that the numbers presented are percentages. Done. See line 800 / Table 2.

Section 3.1.2: This description leaves the reader with several questions, e.g. How many responses were collected at each of the sites? What was the mean and sd? What was indeed the most common frequency of GLV purchase - one, two, or three times a week? What about the remainder 30% of respondents who did not buy GLVs at one outlet? Suggest condensing all this information into one or two tables or figures to make things more transparent. Some interesting findings here about drivers and barriers in GLV consumption. We added a new table (now Table 3) as suggested to report the detailed information on GLV purchase and storage behaviour. 

Line 358: Are there numbers for these? No, unfortunately not, the question on “tips for the Veg-on-wheels” was an open explorative question and handled as qualitative data. 

The discussion does well to include limitations, and the implications section could be strengthened once the methods and results are clearer, with greater reference to evidence on food environments (not systems) from the region and around the world. This is a good suggestion, we added some references on how the food environment shapes food choice, in line with the introduction. See lines 507-509 and 602-606.

Reviewer #2: The manuscript titled A proof-of-principle study with convenience vegetable in urban Nigeria - The Veg-on- Wheels intervention aims at presenting the results of a proof-of-principle study on the purchase of amaranth and fluted pumpkin leaves.

The study focuses on Nigeria, a country where vegetable intake is low due to various constrains ranging from affordability to changes in consumption patterns doe to cultural influences. The study documents well the situation in Nigeria and proposes a sound methodology.

What I found lacking is a clear identification of the research gap to be filled with this proof-of-principle study. To my knowledge, proof-of-principle studies aim at verifying that some concept/ theory/ intervention has practical potential. Agreed, we have not clearly outlined the research gap:

-mobile produce markets have been shown to increase access to fruit and vegetables but have not been studied in LMIC. See lines 121-122.

Workplace seems a promising place to increase access to vegetables but this has not been studied in LMIC. See lines 129-130.

In addition we adjusted the paragraph “A quasi-experiment Veg-on-Wheels” and added the following sentence “Although both mobile markets and interventions a workplaces have been found promising in increasing vegetable access, no such intervention has been conducted in urban areas in LMIC.” See lines 187-189.

Also, in this study we are testing in real-life what is concluded by Hsiao et al. (2019): mobile produce markets are a promising strategy to improve access to fruit and vegetables – and might even support healthy food purchasing and consumption. However, more rigorous experimental designs are needed. However this might not fully cover the term ‘proof-of-principle’ study therefore we adjusted it to ‘explorative intervention study’ in the title and throughout the manuscript (see line 185).

Authors need to stress more how their assessment complements existing knowledge in Nigeria and elsewhere. Also based on suggestions by reviewer 1, we rewrote parts of the introduction, especially former sections 1.2-1.4 where we describe earlier interventions in the food environment for low and middle income countries, earlier studies on consumer determinants, and how this study combines an intervention in the food environment with insights in consumer determinants. 

The study procedure is clear. Authors proposed a quasi-experimental design with control and intervention Periods. Respondents were recruited in the vicinity of the intervention setting sites, near workplaces. For the control period the questionnaire was adapted to match the Nigerian context considered that the food choice motives might differed than what was proposed in previous studies.

The results are presented in detail both in the text and through figures. 

While the section is organized by topics, due to the amounts of information the reader might find it difficult to pin point the take home message. Authors could consider writing a small paragraph at the beginning of the results with the main findings. This is a good suggestion, we added a fewsummarizing sentences at the beginning of the control period results and at the beginning of the intervention results. See lines 377-383 and 441-445.

Small observations

Line 68 – what kind of design innovations? Technological innovations but also organizations innovations. To avoid confusion it was changed to interventions. See lines 138-139. 

Line 91 – what do you mean here by food environment? Based on the comments by reviewer 1, we put more emphasis in the food environment and also added the following: “Food environments have been described as the interface where people interact with the wider food system to acquire and consume foods. It encompasses external (availability, prices, vendor and product properties and marketing and regulation) and personal dimensions (accessibility, affordability, convenience and desirability) (Turner et al., 2018).” See lines 86-90.

Line 104 – Not sure it is a good idea to already draw the conclusions (i.e. Veg-on-Wheels contributed to increased vegetable intake) in the headlines of the introduction Agree, we change the heading to “A quasi-experiment Veg-on-Wheels”. See lines 183-184.

Line 198 – please ad a few lines to explain the purpose of Cronbach`s alphas in this context. We changed the sentence to “Cronbach’s alphas were sufficiently high for all dimensions showing a good interreliability of the scale”, See lines 301-302.

Line 264 – authors need to explain why the sample contained women only. Please do so in the methods section. Indeed, added on lines 248-249: “Respondents were included only if they were responsible for buying GLVs in their households (since in Nigeria this is more often women than men we decided to only include females)”.

Line 357 - see Error! Reference source not found. Corrected, this was the reference to Table 4 (now Table 5) and is now corrected.

---

## [Decision Letter · Decision Letter 1]

15 Jun 2022

PONE-D-21-32704R1An explorative study with convenience vegetables in urban Nigeria - The Veg-on-Wheels interventionPLOS ONE

Dear Dr. Snoek,

Thank you for submitting your manuscript to PLOS ONE. After careful consideration, we feel that it has merit but does not fully meet PLOS ONE’s publication criteria as it currently stands. Therefore, we invite you to submit a revised version of the manuscript that addresses the points raised during the review process.

The manuscript was greatly improved and can be accepted after minor revisions (improving the introduction and discussion section) and a proofreading o manunscript.

We look forward to receiving your revised manuscript.

Kind regards,

Laurentiu Rozylowicz, Ph.D.

Academic Editor

PLOS ONE

Journal Requirements:

Reviewers' comments:

Reviewer's Responses to Questions

**Comments to the Author**

1. If the authors have adequately addressed your comments raised in a previous round of review and you feel that this manuscript is now acceptable for publication, you may indicate that here to bypass the “Comments to the Author” section, enter your conflict of interest statement in the “Confidential to Editor” section, and submit your "Accept" recommendation.

Reviewer #1: (No Response)

Reviewer #2: All comments have been addressed

2. Is the manuscript technically sound, and do the data support the conclusions?

Reviewer #1: Yes

Reviewer #2: Yes

3. Has the statistical analysis been performed appropriately and rigorously? 

Reviewer #1: I Don't Know

Reviewer #2: Yes

4. Have the authors made all data underlying the findings in their manuscript fully available?

Reviewer #1: Yes

Reviewer #2: No

5. Is the manuscript presented in an intelligible fashion and written in standard English?

Reviewer #1: No

Reviewer #2: Yes

6. Review Comments to the Author

Reviewer #1: I thank the authors for their revisions. The manuscript needs some minor editing. Although not essential, I also wonder if effects of respondent characteristics on FCMs can be assessed given the dataset. Specific comments:

In-text references are inconsistent in format - either name all authors, or use et al. after first or second author

The use of the abbreviation LMICs, and the use of commas before and is also inconsistent

L55, L165 Suggest replacing decrease with reduce

L102-106 This sentence needs to correctly use not only ... but also, and could be split into two at also, together, or shape. It is currently incomplete.

L111 food safety (remove hyphen)

L129-131 The word also needs to be replaced between were and included

L134 interventions at workplaces

L179 FUTA and WUR

L194 an additional

L305-313 and L360-365 These summaries and inferences usually go into the Discussion

L308 most consumers

L309 takes a cab(?)

L348 abandoning

Table 3 needs some rewording: Frequency of buying GLV, Number of GLV bundles per purchase (It does not appear that bundles imply diversity), Storage time of fresh/cooked GLV/vegetables, Storage method for fresh/cooked GLV/vegetables

L350-359 Suggest signposting these in FCMs in existing tables/figures, as it is not apparent

L503 It would indeed be interesting to see any effects of education, occupation, and age on the FCMs and vegetable use behaviour

L507-533 As the study deals with the food environment, please replace the term systems with environments throughout

L547 It would be good to know author contributions, and also those of the professors acknowledged here

Table 5 Why is 295 in bold?

Reviewer #2: All my recommendations for the manuscript titled An explorative study with convenience vegetables in urban Nigeria - The Veg-on- Wheels intervention for the have been addressed and answered. The manuscript has clarified its status as proof-of-principle study and is now better situated in the literature addressing food environments. Methodologically, several clarifications have been brought. However, I see several aspects that need further clarification

Introduction:

- the concept of entry point needs some definition/ clarification. Both the food environment (line 59) and consumer behavior (line 123) have been mentioned as entry point in different contexts. Please explain what you mean.

- The section on Vegetable consumption in urban Nigeria ends rather abruptly being followed by more theoretical insights on the food system. To ensure the flow of the introduction, I would suggest moving this section just before the section A quasi-experiment Veg-on-Wheels

Discussions:

- authors mention that the intervention was branded as a FUTA intervention and this has influenced the consumer`s behavior. It would be interesting to discuss how do expect consumer behavior to change in respect to mobile produce markets, not branded as FUTA. How much do you think branding influenced the study outcomes?

-

Line 42: maybe not up to data instead of not current?

7. PLOS authors have the option to publish the peer review history of their article (what does this mean?). If published, this will include your full peer review and any attached files.

Reviewer #1: No

Reviewer #2: No

---

## [Editor Report · Decision Letter 2]

8 Aug 2022

An explorative study with convenience vegetables in urban Nigeria - The Veg-on-Wheels intervention

PONE-D-21-32704R2

Dear Dr. Snoek,

We’re pleased to inform you that your manuscript has been judged scientifically suitable for publication and will be formally accepted for publication once it meets all outstanding technical requirements.

Kind regards,

Laurentiu Rozylowicz, Ph.D.

Academic Editor

PLOS ONE

Additional Editor Comments (optional):

Please note that PLOS One require authors to make all data necessary to replicate their study’s findings publicly available without restriction at the time of publication. Please read https://journals.plos.org/plosone/s/data-availability and be ready to provide additional info to the publisher.
---

## [Editor Report · Acceptance letter]

21 Sep 2022

PONE-D-21-32704R2 

An explorative study with convenience vegetables in urban Nigeria - The Veg-on-Wheels intervention 

Dear Dr. Snoek:

I'm pleased to inform you that your manuscript has been deemed suitable for publication in PLOS ONE. Congratulations! Your manuscript is now with our production department. 

Kind regards, 

on behalf of

Dr. Laurentiu Rozylowicz 

Academic Editor

PLOS ONE